# MYH9 Facilitates Cell Invasion and Radioresistance in Head and Neck Cancer via Modulation of Cellular ROS Levels by Activating the MAPK-Nrf2-GCLC Pathway

**DOI:** 10.3390/cells11182855

**Published:** 2022-09-13

**Authors:** Guo-Rung You, Joseph T. Chang, Yan-Liang Li, Chi-Wei Huang, Yu-Liang Tsai, Kang-Hsing Fan, Chung-Jan Kang, Shiang-Fu Huang, Po-Hung Chang, Ann-Joy Cheng

**Affiliations:** 1Department of Medical Biotechnology and Laboratory Science, College of Medicine, Chang Gung University, Taoyuan 33302, Taiwan; 2Department of Radiation Oncology, Chang Gung Memorial Hospital-Linkou, Taoyuan 33305, Taiwan; 3School of Medicine, College of Medicine, Chang Gung University, Taoyuan 33302, Taiwan; 4Department of Radiation Oncology, New Taipei Municipal TuCheng Hospital, New Taipei City 236017, Taiwan; 5Department of Medical Imaging and Radiological Sciences, College of Medicine, Chang Gung University, Taoyuan 33302, Taiwan; 6Department of Otorhinolaryngology, Chang Gung Memorial Hospital-LinKou, Taoyuan 33305, Taiwan; 7Graduate Institute of Biomedical Sciences, College of Medicine, Chang Gung University, Taoyuan 33302, Taiwan

**Keywords:** MYH9, head and neck cancer, cell invasion, radioresistance, prognosis, reactive oxygen species, GCLC, Nrf2, MAPK signaling pathway

## Abstract

The MYH9 (Myosin heavy chain 9), an architecture component of the actomyosin cytoskeleton, has been reported to be dysregulated in several types of cancers. However, how this molecule contributes to cancer development is still obscure. This study deciphered the molecular function of MYH9 in head and neck cancer (HNC). Cellular methods included clonogenic survival, wound-healing migration, and Matrigel invasion assays. Molecular techniques included RT-qPCR, western blot, luciferase reporter assays, and flow cytometry. Clinical association studies were undertaken by TCGA data mining, Spearman correlation, and Kaplan-Meier survival analysis. We found that MYH9 was overexpressed in tumors and associated with poor prognosis in HNC patients. MYH9 promoted cell motility along with the modulation of the extracellular matrix (fibronectin, ITGA6, fascin, vimentin, MMPs). Also, MYH9 contributed to radioresistance and was related to the expression of anti-apoptotic and DNA repairing molecules (XIAP, MCL1, BCL2L1, ATM, RAD50, and NBN). Mechanically, MYH9 suppressed cellular ROS levels, which were achieved by activating the pan-MAPK signaling molecules (Erk, p38, and JNK), the induction of Nrf2 transcriptional activity, and the up-regulation of antioxidant enzymes (GCLC, GCLM, GPX2). The antioxidant enzyme GCLC was further demonstrated to facilitate cell invasion and radioresistance in HNC cells. Thus, MYH9 exerts malignant functions in HNC by regulating cellular ROS levels via activating the MAPK-Nrf2-GCLC signaling pathway. As MYH9 contributes to radioresistance and metastasis, this molecule may serve as a prognostic biomarker for clinical application. Furthermore, an in vivo study is emergent to support the therapeutic potential of targeting MYH9 to better manage refractory cancers.

## 1. Introduction

Head and neck cancer (HNC) is one of the most prevalent cancers in the world. According to GLOBOCAN 2020 estimates, HNC incidence rises by over 930,000 cases per year, accounting for approximately 5% of all cancers [1]. HNC is a complex disease comprising several anatomic subtypes, including oral cavity, laryngeal, and pharynx cancers [2,3]. Clinically, pathological parameters such as tumor staging, lymph node metastasis, and perineural invasion are well-defined prognostic factors. The standard treatment for HNC includes surgery, radiation, chemotherapy, or a combination of these methods. Although treatment strategies have advanced in recent decades, patient prognosis has not significantly improved. The five-year survival rate of patients with HNC is approximately 80% at the earliest stages, decreasing by over two-fold in the mid-to late-stage [4,5,6].

The head and neck are rich in lymphatic tissues, being the susceptible sites of lymph node metastasis, resulting in the HNC possessing high metastatic potential. However, resistance to radiotherapy leading to cancer recurrence is also a common vulnerability found in HNC [4,5,6]. Interestingly, highly invasive cancers with nodal metastasis are often accompanied by a poor radiotherapeutic response. Similarly, patients with recurrent HNC and radioresistant cancers often have higher metastasis rates [6,7,8]. Recently, an association between cell invasion and radioresistance has been demonstrated. These two malignant attributes share phenotypic crosstalk, and several molecules with both functions have been identified [9]. The molecules participating in these roles may be crucial to the pathological mechanism of HNC. The defined molecules are promising candidates as therapeutic targets for the treatment of HNC.

The myosin heavy chain 9 (MYH9) gene encodes the heavy chain of non-muscle myosin IIA (NMIIA). NMIIA is an actin-binding molecular motor that powers the contraction of the actomyosin cytoskeleton. MYH9 participates in many important cellular processes during development, such as cell cytokinesis, polarization, and adhesion, which are required for maintaining proper cell shape during cell division and migration [10,11]. Although the biological function of MYH9 is well-studied, new findings on diseases related to gene mutations or dysregulation highlight that much remains undiscovered. Recently, the altered expression of MYH9 has been reported in many types of cancer. The overexpression of MYH9 has been found in many cancers, including colorectal, gastric, esophageal, nasopharyngeal, and lung cancers; this upregulation is associated with poor prognosis [12,13,14,15,16,17]. Functionally, MYH9 participates in several malignant processes. Many reports have shown that MYH9 is involved in the facilitation of cancer invasion or metastasis [16,17,18,19,20,21], as well as growth promotion [17,18]. Although the oncogenic function of MYH9 is known in most cancers, there are inconsistencies in reports of its role in tumor suppression. These include a 3D cell culture experiment in lung cancer [22] and mouse model studies in HNC and melanoma [23,24]. Therefore, we hypothesized that MYH9 regulates various oncogenic mechanisms in HNC, which awaits characterization.

In this study, we determined the cellular functions of MYH9 in HNC cells. We found that MYH9 contributed to cell invasion and radioresistance as well. We further demonstrated that these oncogenic functions were achieved by mediating the production of cellular reactive oxygen species via the MAPK-Nrf2-GCLC signaling pathway. Clinically, MYH9 is upregulated in HNC and is associated with a poor prognosis. Our study advances the knowledge regarding the cross-regulatory mechanism of cell invasion and radioresistance; targeting MYH9 may be a promising approach to improve therapeutic effectiveness in HNC.

## 2. Materials and Methods

### 2.1. Cell Lines and Transfection

The head-neck cancer cell lines, including OECM-1 and Detroit 562, were used [25,26]. Both cell lines were purchased from the Taiwan Food Industry Research and Development Institute (Hsinchu, Taiwan). The OECM-1 (also named SCC180) is a human oral squamous cell carcinoma cell line derived from the surgical resection of a primary tumor of a male patient in Taiwan [25]. This cell line contains a genomic mutation in TP53 (https://web.expasy.org/cellosaurus/CVCL_6782, accessed on 1 April 2022) [25]. The Detroit 562 (ATCC cell bank, CCL-138) is a human pharyngeal squamous cell carcinoma cell line derived from the metastatic carcinomatous cells in the pleural fluid of a female Caucasian patient in the USA. The Detroit 562 contains genomic mutations in TP53, CDKN2A, and PIK3CA according to the Sanger COSMIC database (https://web.expasy.org/cellosaurus/CVCL_1171, accessed on 1 April 2022). There is no report of MYH9 mutation in the OECM1 and Detroit 562 cell lines that we used in this study. These cells were maintained in the RPMI-1640 or MEM medium containing 10% FBS at 37 °C in a humidified incubator with 5% CO2 air. For the plasmid transfection, cells were transfected with 6 µg of the specific-shRNA plasmids by Lipofectamine 2000 (Invitrogen, Carlsbad, CA, USA) in Opti-MEM reduced serum media (Invitrogen, Carlsbad, CA, USA). After 12 h, the Opti-MEM medium was replaced with a complete medium.

### 2.2. The Constructions of shRNA Plasmids

The specific short hairpin RNA (shRNA) plasmids were constructed similarly as previously described [9]. In brief, approximately 22 oligonucleotide sense and short antisense hairpin (sh) oligonucleotides were designed to complement the gene-specific mRNA sequence. The gene specific-shRNA was then cloned into the pGSH1-GFP vector plasmid. The sequence for shMYH9 was 5′-GAT-CCG-CCA-AGC-TCA-AGA-ACA-AGC-ATG-AAG-CTT-GAT-GCT-TGT-TCT-TGA-GCT-TGG-CTT-TTT-TGG-AAG-C-3′.

### 2.3. Analyses of Cell Migration

The cell migration was examined by an in vitro wound-healing assay similarly, as previously described [27]. Briefly, cells were seeded in an ibidi^®^ culture insert (ibidi LLC, Verona, WI, USA) on top of a six-well plate. After 8 h of incubation, the culture insert was detached to form a cell-free gap in a monolayer of cells. After changing to culture medium with 1% FBS, the cell migration status toward the gap area was photographed in periods.

### 2.4. Analyses of Cell Invasion

Cell invasion ability was evaluated by using a BioCoat Matrigel (Becton Dickinson Biosciences, Bedford, MA, USA) and Millicell invasion chamber (Millipore Corporation, Bedford, MA, USA) [26,27]. The Matrigel was first coated onto the membrane of the Millicell upper chamber with a pore size of 8 μm in a 24-well plate. Cells in 1% FBS medium were seeded into the upper chamber. The lower chamber will contained 10% FBS in medium to trap invading cells. The invasion ability was determined by observing the reverse side of the upper chamber after being fixed and stained with crystal violet.

### 2.5. Assessment of Radiosensitivity

Radiosensitivity was determined by clonogenic survival, as previously described [26,27]. Briefly, cells were seeded into a six-well cell culture plate for 8 h. The cells were then exposed to radiation (0 to 6 Gy) and continuously cultured for seven–14 days to allow cell colony formation. The tested cells were stained with crystal violet, and the survival fraction was calculated as the number of colonies divided by the number of seeded cells times the plating efficiency.

### 2.6. RT-qPCR Method to Determine mRNA Expressions

RNA extraction and cDNA synthesis were performed as previously described [27]. Briefly, the PCR and cDNA synthesis was performed on a MiniOpticon™ real-time PCR detection system using SyBr Green Supermix reagents. The primers used in this study are listed in Appendix A.

### 2.7. Western Blot Method to Determine Protein Levels

As previously described, the cellular protein extraction and western blot assay were performed [26]. Briefly, cells were homogenized in a lysis buffer, incubated on ice for 30 min, and centrifuged to obtain the cellular proteins. For western blot analysis, the protein extract was subjected to SDS-polyacrylamide gel for electrophoresis. After transferring the protein image, the membrane was incubated to a specific primary-antibody and second-antibody conjugated with horseradish peroxidase. The membrane was treated with ECL developing solution and exposed to X-ray film. For each assay, the GAPDH expression was used as an internal control. The primary antibodies used in this study are listed in Appendix A.

### 2.8. Measurement of Cellular Reactive Oxygen Species (ROS) Level

Cellular ROS level was measured in live cells by the H2DCF-DA oxidation method (Invitrogen, Carlsbad, CA, USA) similarly as previously described [27]. Briefly, the test cells were suspended in a PBS buffer supplemented with an H2DCF-DA reagent. The H2DCF-DA is a cell-permeable probe oxidized by intracellular ROS to generate fluorescent DCF. The fluorescence intensity of DCF was then determined by flow cytometric analysis (FACSCalibur, BD Biosciences, Franklin Lakes, NJ, USA). For normalization, we used the relative average intensity of the living cells (PI-negative) in 10,000 cells via flow cytometric measurement. There were approximately 90% of living cells in all treatments.

### 2.9. Luciferase Report Assay for the Transcriptional Activity of Nrf2

The plasmid of the pGL3 promoter-8xARE luciferase reporter construct purchased from MDbio Inc. (Taipei, Taiwan) was used to assess Nrf2 transcriptional activity. This construct contains eight copies of antioxidant response elements [ARE, 5′-GTGACAAAGCA-3′] in the promoter of the luciferase reporter gene that can be driven by ARE when Nrf2 binds to activate it [28]. This reporter plasmid was transfected to HNC cells to evaluate Nrf2 transcriptional activity by determining luciferase activity. After 24 h of transfection, the luciferase activity was determined using the Steady-Glo luciferase assay system (Promega, WI, USA), and the luciferin levels were measured by GloMax^®^ 20/20 Luminometer. The relative activity was presented by the fold-change after comparison to the level in the control cells.

### 2.10. Clinical Association and Prognostic Evaluation of MYH9 in HNC Patients

The Cancer Genome Atlas (TCGA)-HNSC cohort was used for clinical analysis. The GEPIA2 platform (http://gepia2.cancer-pku.cn/#index, accessed on 7 January 2022) [29] was applied to compare the transcript levels of each gene between HNC tumors (*N* = 519) and the normal tissues (*N* = 44). A Spearman correlation analysis was performed to evaluate the relationship between MYH9 and the interest genes in the TCGA-HNSC dataset. The KM-Plotter online tool (http://kmplot.com/analysis, accessed on 12 April 2022) was used to determine the prognostic significance of MYH9 in HNC patients (*N* = 500). High- and low-risk groups were classified using an optimization algorithm in the order of the prognostic index according to each gene expression level by the “auto select best cutoff” setting. A Kaplan-Meier analysis was performed to evaluate overall survival. The hazard ratios (HRs) with 95% confidence intervals (CIs) were calculated using the log-rank test.

## 3. Results

### 3.1. MYH9 Promotes Cell Motility along with the Modulation of the Extracellular Matrix

We first analyzed the basal levels of MYH9 between 5 HNC cell lines and two lines of normal oral keratinocytes. As shown in Figure 1A, MYH9 was significantly overexpressed in the HNC cell lines, with an average elevation of two-fold. Two HNC cell lines, the OECM1 from the oral cavity and the Detroit line from the oropharynx, were selected because they represent the commonality of cancer sites often found in HNC. The cellular motility function of MYH9 was examined via the MYH9-silencing approach. Figure 1B shows the typical result of the efficacy of MYH9 knockdown. In each experiment, the shMYH9 knockdown efficacy was confirmed to be greater than 70% before functional and molecular analyses. The cell migration and invasion were assessed using in vitro wound healing and Matrigel invasion assays. As shown in Figure 1C, cell migration was significantly inhibited by the gene knockdown. At 24 h, the migration ability of OECM-1 and Detroit 562 cells were reduced to 51% and 40%, respectively. Cell invasion was consistently suppressed after 24 h of gene knockdown, decreasing to 32% and 47% in OECM-1 and Detroit 562 cells, respectively (Figure 1D). Thus, MYH9 silencing reduces the ability of HNC cells to migrate and invade.

To determine whether MYH9 function is related to molecular presentation in clinical cancers, we investigated the potential association of MYH9 expression levels with motility-related molecules using a TCGA-HNSC dataset. Extracellular matrix (ECM) molecules and metalloproteinase (MMPs) were examined, including fibronectin (FN1), integrin-α6 (ITGA6), fascin (FSCN1), vimentin (VIM), MMP2, and MMP9. The expression levels of these proteins are shown in Figure 1E. As shown, all these proteins were significantly upregulated in tumors compared with normal tissues (*p* < 0.001 for all proteins). Furthermore, in tumor tissues, the expression levels of these genes were all correlated with MYH9 expression (Figure 1F). These results indicate that the function of MYH9 in cell motility may occur via the regulation of ECM integrity.

### 3.2. MYH9 Contributes to Radioresistance and Is Associated with the Anti-Apoptotic Mechanism

The potential effect of MYH9 on radiosensitivity was examined using clonogenic survival assays (Figure 2). In the two tested cell lines, MYH9 silencing increased cell death in response to irradiation. Compared to the control cells, the number of surviving colonies was reduced by 21% at 6 Gy in OECM-1 cells (Figure 2A) and by 40% at 4 Gy in Detroit 562 cells (Figure 2B). Thus, MYH9 silencing decreases radioresistance and sensitizes HNC cells to irradiation.

Furthermore, we determined the potential association of MYH9 with the clinical presentation of anti-apoptotic and DNA repair molecules using the TCGA-HNSC dataset. For anti-apoptotic molecules, we evaluated the X-linked inhibitor of apoptosis protein (XIAP), myeloid cell leukemia-1 (MCL1), and BCL-2-like protein 1 (BCL2L1). For DNA repair molecules, we examined the radiation-induced pathway of the ATM-MRN complex (RAD50, NBN) [30]. The expression levels of these proteins are shown in Figure 2C. These anti-apoptotic proteins (XIAP, MCL1, and BCL2L1) and DNA repair genes (ATM, RAD50, and NBN) were significantly upregulated in tumors compared to normal tissues. In addition, the expression levels of these proteins were significantly correlated with those of MYH9 (Figure 2D). These results indicated that MYH9 contributes to radioresistance in HNC and may occur via increasing anti-apoptotic and DNA repair effects to gain a survival advantage.

### 3.3. MYH9 Suppressed Cellular ROS Levels via Activation of Nrf2 and Up-Regulation of Antioxidant Enzymes

It is well established that ionizing radiation induces the production of reactive oxygen species (ROS) in cells, leading to apoptosis. Therefore, we examined whether MYH9 exerts an effect on the regulation of cellular ROS levels. Intracellular ROS were investigated using the H2DCF-DA oxidation method, and the green fluorescence DCF product was analyzed using flow cytometry [26]. As shown in Figure 3A, MYH9 silencing increased intracellular ROS production in both of the HNC cell lines. Compared to the control cells, the ROS level increased by two-fold in OECM-1 cells and 1.3-fold in Detroit 562 cells. These results suggest that MYH9 plays a role in the suppression of ROS generation.

To further investigate the molecular mechanism of MYH9 in ROS modulation, we used a luciferase reporter assay with an antioxidant response element (ARE), to assess the well-known Nrf2-inducing transcriptional activity of antioxidant enzymes [28]. As shown in Figure 3B, MYH9 silencing significantly repressed ARE-luciferase activity in the two HNC cell lines. Compared with the controls, Nrf2 transcriptional activity was reduced by 30 and 58% in OECM-1 and Detroit 562 cells, respectively.

We further examined the effect of shMYH9 on the mRNA expression of six Nrf2 antioxidant enzymes in HNC cells, including glutamate cysteine ligase catalytic subunit (GCLC), glutamate cysteine ligase modifier subunit (GCLM), glutathione peroxidase 2 (GPX2), glutathione synthetase (GSS), manganese superoxide dismutase (MnSOD), and catalase. As shown in Figure 3C, GCLC, GCLM, and GPX2 were substantially reduced following MYH9 silencing. To confirm this MYH9 modulatory effect, we examined the protein levels of these three enzymes. As shown in Figure 3D, these ROS scavengers were consistently inhibited in response to MYH9 silencing. In the most prominent molecule, GCLC, the protein level decreased to 36 and 51% in the two HNC cell lines compared to the controls. These results demonstrate that MYH9 suppressed cellular ROS levels by the induction of Nrf2 activity to up-regulate the expression of the antioxidant enzymes GCLC, GCLM, and GPX2.

To manifest this cellular finding in a clinical presentation, we examined the expression levels of these enzymes in patients with HNC using the TCGA-HNSC dataset. As shown in Figure 3E, although there was no significant correlation between GPX2 and MYH9 in clinical presentation, GCLC and GCLM were significantly upregulated in tumors, and their molecular levels were statistically correlated with MYH9 expression (*p* < 0.001 in GCLC-MYH9 and GCLM-MYH9). Taken together, MYH9 contributes to radioresistance that may occur through the elevation of the antioxidant enzymes, reducing cellular ROS levels, and resulting in apoptotic resistance. MYH9 silencing may be an integral approach to balancing ROS status and reversing this radioresistant effect.

### 3.4. MYH9 Activates Pan-MAPK Signaling Pathways Leading to ROS Dysregulation

Several signaling pathways are commonly altered in cancer, leading to dysregulated cellular functions and cancer progression. These oncogenic signaling molecules include PI3K-PDK1/Akt, MAPK-Jnk/p38/Erk, and Wnt/GSK3b, which modulate several cell processes, including proliferation, survival, and invasion [31,32,33,34,35]. We investigated whether MYH9 induces malignant phenotypes related to these signaling pathways. To this end, alterations in the phosphorylation status of these signaling molecules were examined using western blotting analysis following MYH9 silencing. As shown in Figure 4A, MYH9 exerted minimal effects on pGSK3b, pPDK1, and pAkt levels. However, the phosphorylated states of MAPK family proteins, including pErk (MAPK3), pP38 (MAPK1), and pJNK (MAPK8), were significantly reduced, decreasing to 46 and 55% of the expression level of control factors upon MYH9 silencing (Figure 4B). Since the p-GSK3b plays a crucial role in the activated Wnt signaling pathway, the minimal alteration of this molecule suggested an insignificant effect of MYH9 in modulating the Wnt signaling pathway. Thus, these results suggest that MYH9 induces many MAPK signaling pathways by stimulating the activity of multiple kinases.

To determine whether MAPK phosphorylation affects the expression of ROS scavengers, we employed an Erk activator (ceramide C6) and a p38/JNK activator (anisomycin), which we found to increase pErk and pJNK expression, respectively (Figure 4C). These findings are concurrent with the elevated expression of the ROS scavengers GCLC, GCLM, and GPX2. These results suggest that these antioxidant enzymes are downstream proteins regulated by the MAPK signaling pathway.

To confirm whether the modulation of antioxidant enzymes by MYH9 is associated with MAPK, we examined GCLC/GCLM expression levels in response to MYH9 silencing while maintaining MAPK activity by treating cells with specific activators (Figure 4D). In general, GCLC and GCLM exhibited similar responses; they were inhibited by MYH9 silencing and eliminated by Erk (ceramide C6) and p38/JNK (anisomycin) induction. However, the suppressive effects on GCLC/GCLM of MYH9 silencing were abolished by MAPK activation. These results suggest that MYH9 regulates cellular ROS production through the MAPK-GCLC/GCLM molecular pathway. These cellular findings are supported by clinical presentation, as analyzed in the TCGA-HNSC dataset. As shown in Figure 4E, p38 and JNK were upregulated in tumors compared to normal tissues, and the expression of these MAPK proteins were correlated with MYH9 levels (*p* < 0.001 in pMYH9-p38 and MYH9-JNK).

### 3.5. GCLC Modulated by MYH9 Facilitates Cell Invasion and Radioresistance

To further link the molecular axis of MYH9–GCLC with cellular function, we examined the effects of GCLC on the regulation of cell invasion and radioresistance. GCLC expression was knocked down by transfecting GCLC-specific shRNA (shGCLC) into two HNC lines. As shown in Figure 5A, GCLC silencing considerably suppresses cell invasion by 50 and 69% in OECM-1 and Detroit 562 cells, respectively. The effect of GCLC on radiosensitivity is shown in Figure 5B. GCLC silencing significantly reduced radioresistance to 22 and 43% of the control level in OECM-1 and Detroit 562 cells, respectively. Thus, the dual function of MYH9 in cell invasion and radioresistance may result from the modulation of intracellular ROS levels by facilitating GCLC expression. Nevertheless, the possibility of GCLC in modulation MYH9 may not be excluded.

### 3.6. MYH9 Is Overexpressed in Tumors and Associated with Poor Prognosis in Patients with HNC

To determine whether the malignant characteristics of MYH9 found in cell invasion and radioresistance are reflected in the clinical presentation of patients with HNC, we investigated the differential expression of MYH9 in normal and cancer tissues using the TCGA-HNSC dataset. The patient characteristics and clinical information, including age, gender, tumor sites, and clinical stage, were summarized in Appendix A. As shown in Figure 6A, MYH9 was overexpressed in tumors (*p* < 0.0001), supporting the oncogenic function of this molecule in cancer formation. To extend the effect of MYH9 on prognosis, we examined the association between patient survival and gene expression levels in the TCGA-HNSC cohort (*N* = 500). As shown in Figure 6B, high levels of MYH9 were significantly correlated with poor survival (*p* = 0.045, HR = 1.32). These results suggest that MYH9 functions in cancer aggressiveness and MYH9 may be a crucial molecule as a prognostic biomarker and therapeutic target for patients with HNC.

Taken together, this study deciphered that MYH9 leads to HNC through exerting the motility and radioresistant functions by regulating cellular ROS levels via activating the MAPK-Nrf2-GCLC signaling pathway (Figure 6C). Nevertheless, a more in-depth investigation is warranted to assess all sites of HNC before further clinical application of this knowledge.

## 4. Discussion

With growing insights into molecular alterations that may drive cancer progression, the identification of critical molecules participating in oncogenic processes is imperative and may uncover effective therapeutic targets. This study revealed that MYH9 expression is a crucial factor in HNC, which can be summarized in a few points. (1) MYH9 is overexpressed in tumors and associated with poor prognosis in patients with HNC. (2) MYH9 promotes cell motility, which is associated with modulation of the extracellular matrix. MYH9 silencing reduces the migration and invasion of HNC cells. (3) MYH9 contributes to radioresistance and is associated with an anti-apoptotic mechanism. MYH9 silencing sensitizes radiation response in HNC cells. (4) MYH9 reduces cellular ROS levels. This effect is achieved by activating MAPK signaling, Nrf2 transcription, and anti-oxidant enzymes (GCLC, GCLM, GPX2). (5) The antioxidant enzyme GCLC, which is modulated by MYH9, facilitates cell invasion and radioresistance. GCLC silencing diminishes these malignant attributes in HNC cells.

As MYH9 encodes the heavy chain of NMIIA, an architectural component of the actomyosin cytoskeleton [10,11], dysregulation of this gene may lead to cell invasion and cancer metastasis. Previously, MYH9 has been reported to inhibit cell invasion, presuming (through the modulation of TP53 nuclear retention) to affect p53-responsive genes [36]. In this study, we demonstrated that MYH9 facilitates cell invasion in HNC (Figure 1), and it has been shown in other cancers as well [16,17,18,19,20,21]. Several cellular mechanisms may be involved, including the modulation of epithelial-mesenchymal transition [18,19], focal adhesion assembly [20], and extracellular matrix (Figure 1D). Recent reports of MYH9 in other malignant functions have expanded the significance of this molecule. MYH9 may promote cell growth in colorectal cancer [17,18], induce stem-like properties in lung cancer [15], and reduce chemosensitivity in gliomas and colorectal cancer [17,21]. These effects may relate to regulating stemness-associated markers such as CD44 and CD133 [15,17]. In this study, we observed that MYH9 has a novel oncogenic function. We first showed that MYH9 contributes to radioresistance in cancer, accompanied by the regulation of antiapoptotic molecules (Figure 2C). Moreover, MYH9 reduced the intracellular ROS levels (Figure 3A). This lower ROS level may increase cell resistance to irradiation and resistance to other chemotherapeutic drugs [17,21].

In addition to the function of architecture in assembling actomyosin filaments, MYH9 may participate in signaling transduction pathways. MYH9 contains an IQ domain and head-like myosin domain. These domains have several phosphorylation sites for the corresponding kinases that are thought to acquire motor power by obtaining ATP for contractility [10,11]. Aside from its downstream position being phosphorylated, studies showed that MYH9 might also serve as an upstream molecule to activate several oncogenic signaling pathways. In glioma and pancreatic cancer, MYH9 may interact with GSK3β, subsequently triggering the Wnt/β-catenin signaling pathway to facilitate cancer progression [19,21]. MYH9 may activate mTOR signaling in lung and colorectal cancers to induce cancer stem-like or chemo-resistant phenotypes [15,17]. In glioma and colorectal cancer, MYH9 may induce PI3-Akt/Akt signaling to facilitate cancer growth and metastasis [18,21]. In this study, we did not observe dysregulation in the GSK3β, PDK1, or Akt pathways in HNC (Figure 4). Instead, MYH9 activated pan-MAPK signaling pathways, including Erk, p38, and JNK (Figure 4). This finding is supported by reports of the MYH9-MAPK signaling axis in colorectal cancer [18]. Taken together, MYH9 regulates multiple malignant phenotypes via various pathways, perhaps in a cancer type-specific manner.

ROS generation is a critical factor in cells exposed to irradiation, leading to apoptosis [26]. By analyzing how MYH9 contributes to radioresistance in HNC, we found that this molecule plays a role in reducing cellular ROS levels (Figure 3A). We further investigated the molecular mechanism by which MYH9 regulates redox homeostasis. Nrf2 is an emerging signaling molecule of a transcriptional factor that balances oxidative stress in cells. Under normal conditions, Nrf2 is ubiquitinated through KEAP1 and degraded by the proteasome [37,38]. Following exposure to electrophiles or oxidative stress, KEAP1 is inactivated. The stabilized Nrf2 translocates into the nucleus and induces the expression of a wide variety of detoxification and antioxidant enzyme genes [37,38]. Although Nrf2 may play a dual role in cancer, acting as an oncogene or tumor suppressor [37,38], studies have indicated that constitutive activation of Nrf2 may deliver selective advantages to cells that promote their adaptation to adverse growth conditions for cancer formation [39,40,41]. Nrf2 is associated with cancer invasion, chemoresistance, radioresistance, and poor clinical prognosis in many cancers [41,42,43,44]. Furthermore, Nrf2 is activated by various oncogenic kinases, such as GSK3β, PI3-AKT, and MAPK [45]. Our findings support the oncogenic function of Nrf2 and suggest that MYH9 modulates Nrf2 induction. Silencing MYH9 inhibits Nrf2/ARE signaling, leading to modulation of cellular ROS levels (Figure 3B).

Antioxidant enzymes play a significant role in cellular defense against ROS. Glutathione, a primary cellular non-protein antioxidant and redox regulator, has been reported to be altered in patients with many types of cancer [46]. Several antioxidant enzymes that are driven by Nrf2 participate in glutathione metabolism, including GCLC, GCLM, GSS, and GPX2 [36,46]. To identify the downstream effectors of Nrf2, we examined the transcriptional levels of these antioxidant enzymes in response to MYH9 modulation. We found that MYH9 silencing considerably reduced the expression of GCLC, GCLM, and GPX2 at both the mRNA and protein levels (Figure 3C,D), suggesting that MYH9 regulates cellular ROS levels via the Nrf2-GCLC/GCLM/GPX2 molecular axis. The clinical presentation supported these findings that GCLC and GCLM are significantly upregulated in tumors, and their expression levels are statistically correlated with MYH9 expression (Figure 3E). Because MYH9 taking part in the redox regulation is upstream and via an indirect mechanism, the clinical correlations of MYH9 to the immediate effectors of the antioxidant enzymes (GCLC/GCLM), though statistically significant, may not be very high. However, we showed the causal relationship between these molecules through molecular rescue experiments. We noted that the suppressive effects of GCLC/GCLM on MYH9 silencing were abolished by MAPK activation (Figure 4D). These results demonstrate that MYH9 regulates cellular ROS through the MAPK-Nrf2-GCLC/GCLM signaling pathway. Recently, dysregulation of glutamate-cysteine ligases has been reported to be associated with cancer aggressiveness. For example, a high level of GCLC is associated with poor prognosis in patients with acute myeloid leukemia and hepatocellular carcinoma [47,48]. GCLC and GCLM confer chemoresistance in lung, breast, and head and neck cancers [49,50,51,52]. In this study, we showed that GCLC contributes to cell invasion and radioresistance in HNC (Figure 5). To the best of our knowledge, this is the first report on the direct function of GCLC in these malignant attributes.

## 5. Conclusions

Metastasis and radioresistance are important causes of treatment failure in patients with HNC. As MYH9 contributes to these attributes, this molecule is a great candidate for clinical applications as a prognostic or predictive biomarker. In the future, an in vivo study will be undertaken to support the therapeutic potential of targeting MYH9 for refractory cancer.

## Figures and Tables

**Figure 1 cells-11-02855-f001:**
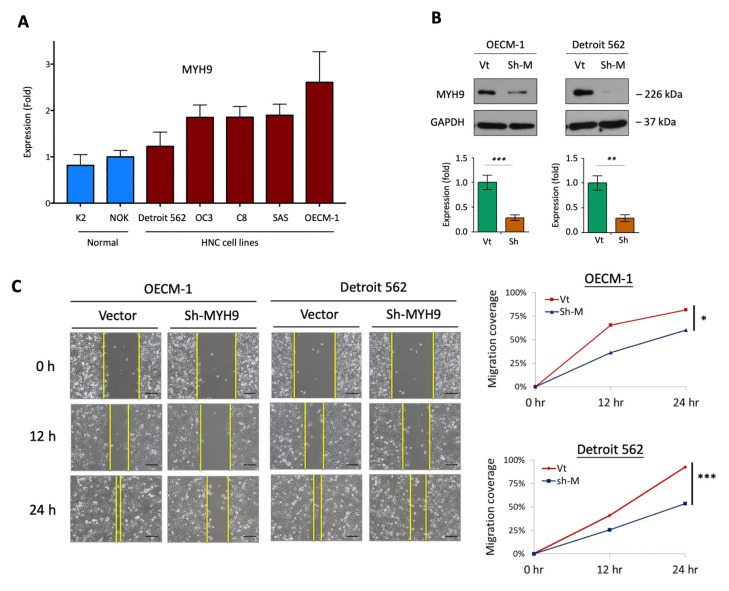
MYH9 promoted cell motility in HNC cells. (**A**) Examination of MYH9 expression level in five HNC cell lines (Detroit 562, BM2, OC3, C8, SAS, OECM-1) and two lines of keratinocytes (K2 and NOK) using the RT-qPCR method. (**B**) The efficacy of MYH9-shRNA knockdown in OECM-1 and Detroit 562 cells. The MYH9 protein level was determined by the western blot method to show the effect of MYH9 silencing. The GAPDH level was used as an internal control. (**C**) Transfection of MYH9-shRNA significantly decreased cell migration, as determined by an in vitro wound healing assay, scale bar 40 μm. (**D**) Transfection of MYH9-shRNA suppressed cell invasion, as determined by a Matrigel invasion assay, scale bar 20 μm. (**E**) Significant increases of ECM/MMPs-associated gene expressions in the tumor samples compared to normal samples. The gene expression data, including FN1, ITGA6, FSCN1, VIM, MMP2, and MMP9, were retrieved from the TCGA-HNSC dataset. (**F**) Correlative expressions between MYH9 and ECM/MMPs associated molecules FN1, ITGA6, FSCN1, VIM, MMP2, and MMP9 in the HNC tumor samples. The gene expression data were retrieved from the TCGA-HNSC dataset. All of the experiments were performed at least three times independently, and a typical result was shown. The error bars shown in the relevant figures indicated the standard deviation of the quantification results in all experiments. (*: *p* < 0.05, **: *p* < 0.01, ***: *p* < 0.001, *t*-test).

**Figure 2 cells-11-02855-f002:**
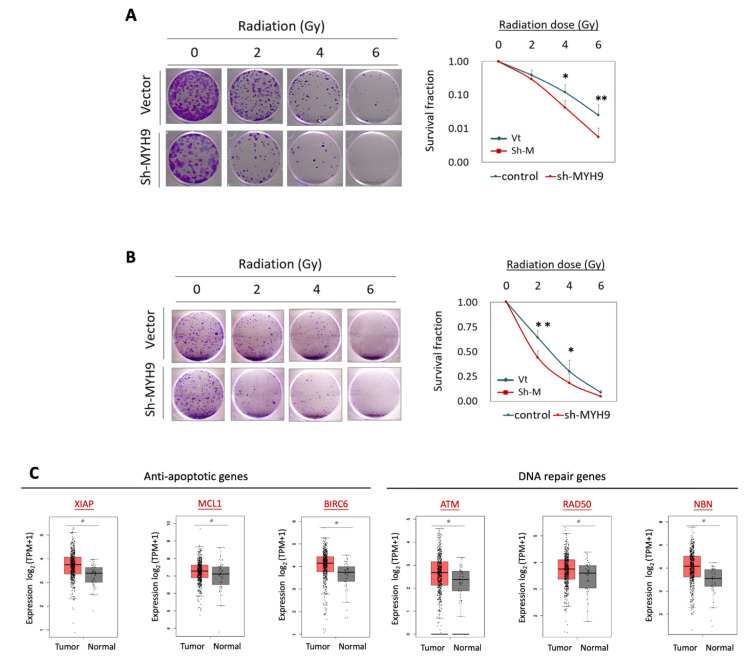
MYH9 contributed to radioresistance in cancer cells. (**A**,**B**) Transfection of MYH9-shRNA increased radiosensitivity in OECM-1 (**A**) and Detroit 562 (**B**) cells, as determined by clonogenic survival assay. (**C**) Significant increases of anti-apoptotic and DNA repair-associated gene expressions in the tumor samples compared to normal samples. The gene expression data, including XIAP, MCL1, BCL2L1, ATM, RAD50, and NBN, were retrieved from the TCGA-HNSC dataset. (**D**) Correlative expressions between MYH9 and anti-apoptotic and DNA repair associated molecules XIAP, MCL1, BCL2L1, ATM, RAD50, and NBN, in the HNC tumor samples. The gene expression data were retrieved from the TCGA-HNSC dataset. All the experiments were performed at least three times independently, and a typical result was shown. The error bars shown in the relevant figures indicated the standard deviation of the quantification results in all experiments. (*: *p* < 0.05, **: *p* < 0.01, *t*-test).

**Figure 3 cells-11-02855-f003:**
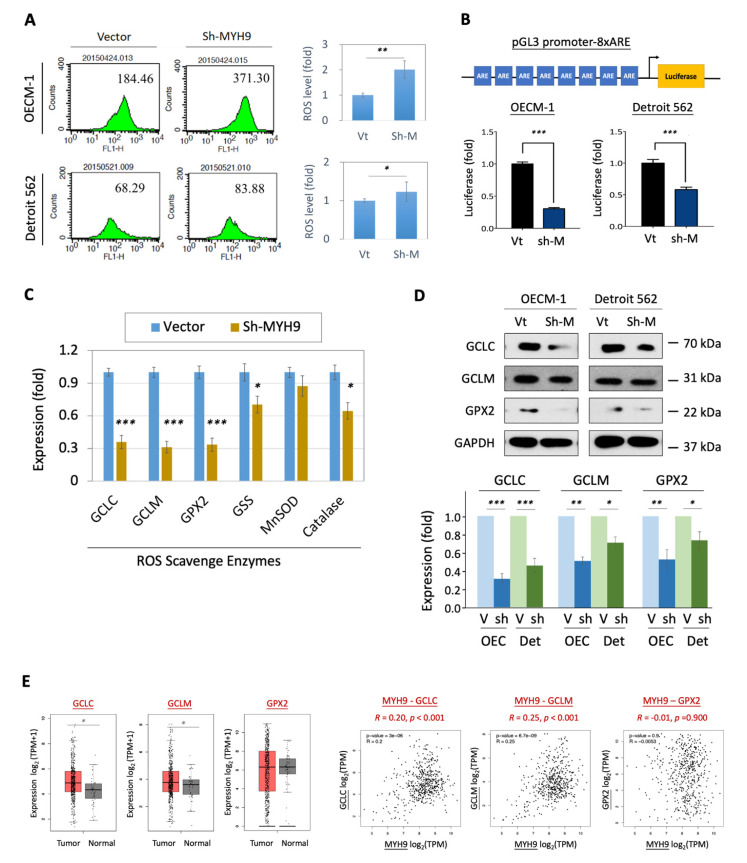
MYH9 modulated intracellular ROS level leads to radioresistance in HNC cells. (**A**) MYH9 silencing increased ROS production in HNC cells. After transfection of MYH9-shRNA plasmids in theboth HNC cells (OECM-1 and Detroit 562), the ROS level was determined using the H2DCF-DA oxidation method and analyzed by flow cytometry. (**B**) MYH9 silencing reduced the Nrf2 transcriptional activity. Upper: A schematic representation of the pGL3 promoter-8xARE construct which was identified for measuring Nrf2 transcriptional activity. Lower: A luciferase report assay was used to compare the relative level of Nrf2 transcriptional activity in the HNC cells (OECM-1 and Detroit 562) transfected with MYH9-shRNA (sh-M) versus control transfectant (Vt). (**C**,**D**) MYH9 silencing inhibited the expression of ROS scavenger enzymes at the mRNA level of the OECM1 cells (**C**) and the protein (**D**) levels in two HNC cell lines. (**E**) Significantly higher expressions of ROS scavenger enzymes (GCLC, and GCLM) in HNC tumor samples compared to the normal samples. Correlative expressions between MYH9 and ROS scavenger enzymes-associated molecules GCLC and GCLM, in the HNC tumor samples. The gene expression data were retrieved from the TCGA-HNSC dataset. All the experiments were performed at least three times independently, and a typical result was shown. The error bars shown in the relevant figures indicate the standard deviation of the quantification results in all experiments. (*: *p* < 0.05, **: *p* < 0.01, ***: *p* < 0.001, *t*-test).

**Figure 4 cells-11-02855-f004:**
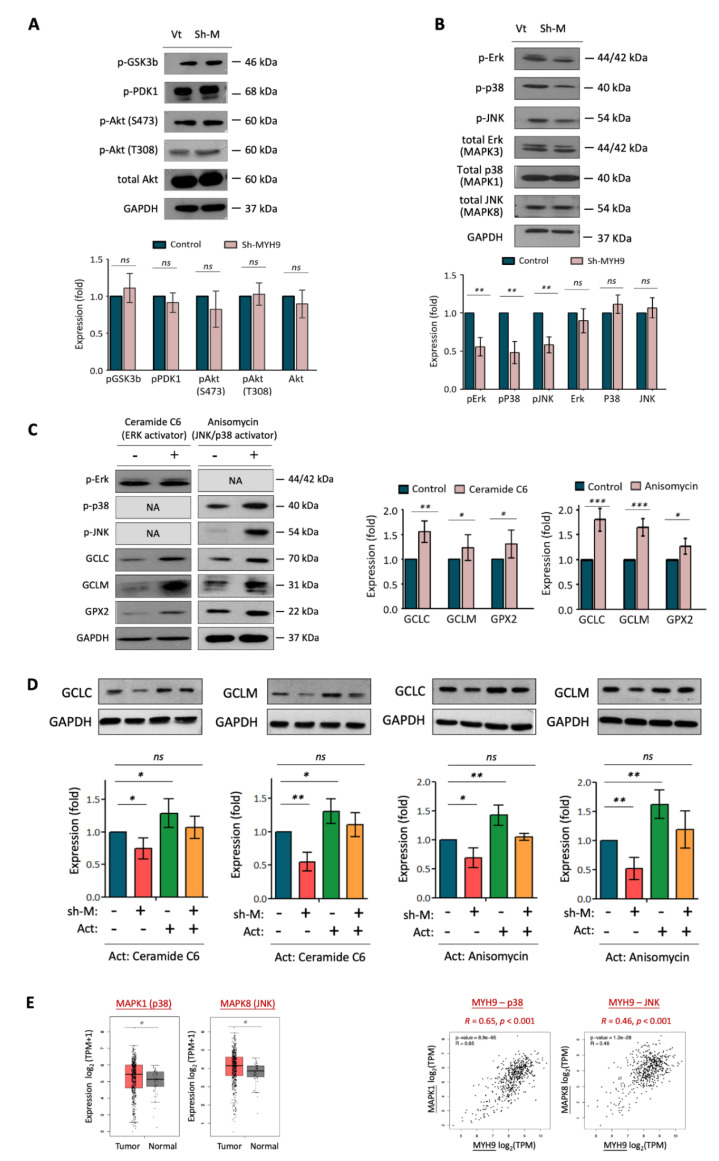
MYH9 induces the pan-MAPK signaling pathway. (**A**) MYH9 silencing had no influence on the PI3K-PDK1/Akt and GSK3β pathway in OECM-1 cells. After transfection of the MYH9-shRNA (sh-M) or the vector (Vt) plasmid into OECM-1 cells, cellular proteins were extracted and subjected to western blot analysis for pGSK3b, pPDK1, pAkt (S473), pAkt (T308), and total Akt protein expressions. (**B**). MYH9 silencing inhibits the phosphorylated status of MAPK signaling molecules. After transfection of the MYH9-shRNA (sh-M) or the vector (Vt) plasmid into OECM-1 cells, cellular proteins were extracted and subjected to western blot analysis for phosphorylated and total levels of Erk/p38/JNK. (**C**) Activators of Erk and p38/Jun activate ROS scavenger enzymes (GCLC/GCLM). After treatment of the Erk activator (Ceramide C6) or p38/JNK (Anisomycin) in OECM-1 cells, the phosphorylated status of Erk/p38/JNK and ROS scavenger enzymes, GCLC, GCLM, and GPX2 were determined by western blot analysis. (**D**) The effect of GCLC/GCLM by shMYH9 was abolished upon MAPK activation. After treatment of Erk activator (Ceramide C6) or p38/JNK (Anisomycin) in the vector control or shMYH9 silencing cells, the protein GCLC and GCLM were determined by western blot analysis. (**E**) Significantly higher expressions of MAPK1 and MAPK8 in HNC tumor samples compared to the normal samples. Correlative expressions between MYH9 and MAPK1 or MAPK8 in the HNC tumor samples. The gene expression data were retrieved from the TCGA-HNSC dataset. The GAPDH level was used as an internal control. All the experiments were performed at least three times independently, and a typical result was shown. The error bars shown in the relevant figures indicated the standard deviation of the quantification results in all experiments. (*: *p* < 0.05, **: *p* < 0.01, ***: *p* < 0.001, ns: no significance, *t*-test).

**Figure 5 cells-11-02855-f005:**
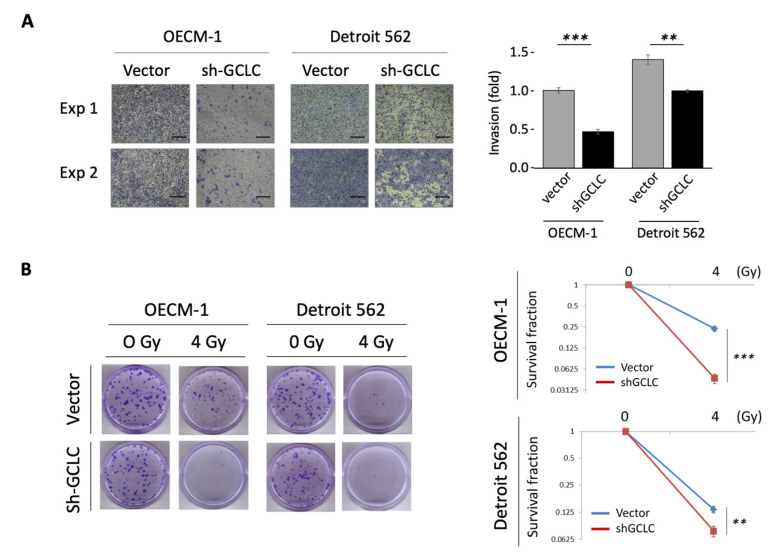
GCLC increased cell motility and radioresistance in HNC cells. (**A**) Transfection of GCLC-shRNA suppressed cell invasion OECM-1 and Detroit 562 cells, as determined by Matrigel invasion assay, scale bar 40 μm. (**B**) Transfection of GCLC -shRNA increased radiosensitivity in OECM-1 and Detroit 562 cells, as determined by clonogenic survival assay. All the experiments were performed at least three times independently, and a typical result was shown. The error bars shown in the relevant figures indicated the standard deviation of the quantification results in all experiments. (**: *p* < 0.01, ***: *p* < 0.001, *t*-test).

**Figure 6 cells-11-02855-f006:**
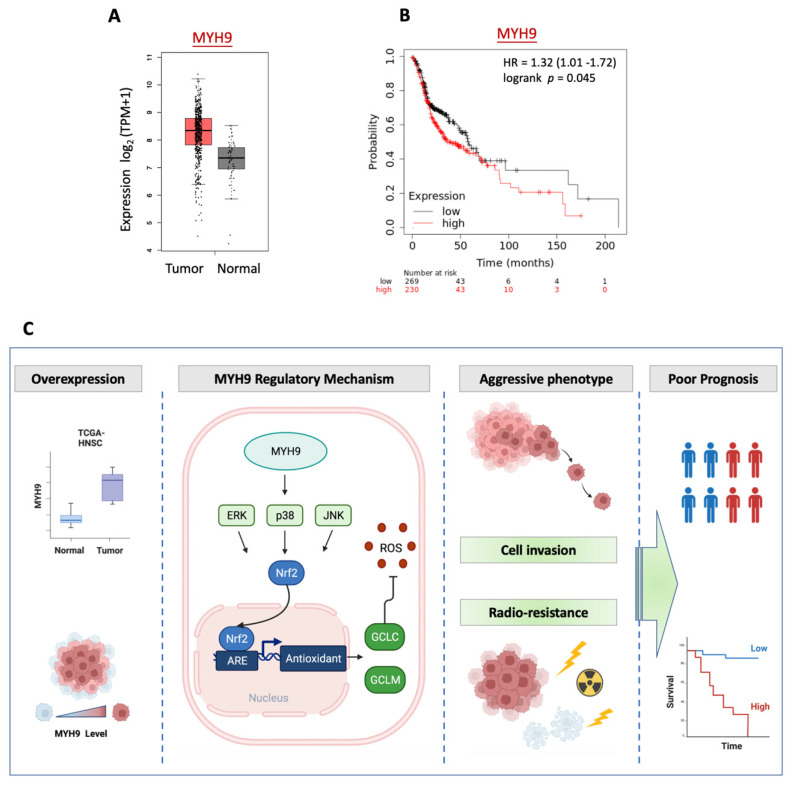
Clinical association of MYH9 in HNC patients. (**A**) Differential expression of MYH9 in HNC and normal tissues, as determined using TCGA-HNSC datasets. (**B**) Prognostic significance of MYH9 in HNC patients, as determined using the TCGA-HNSC dataset by the KM plotter method (*n* = 500). The survival curve, hazard ratio (HR), and *p*-value are indicated. (**C**) A molecular model representing MYH9 regulatory mechanism resulting in cell invasion and radioresistance in HNC cells. MYH9 activates the pan-MAPK signaling molecules, including Erk, p38, and JNK. This activation leads to the induction of Nrf2 transcriptional activity, the upregulation of the antioxidant enzymes, and the reduction of cellular ROS levels. The antioxidant enzyme, such as GCLC, further confers cell invasion and radioresistance, resulting in aggressive cancer and poor prognosis. Taken together, MYH9 exerts malignant effects in HNC via regulating cellular ROS levels by activating the MAPK-Nrf2-GCLC signaling pathway.

## Data Availability

The data of TCGA-HNC patients used in this study are from the publicly accessible GEPIA2 and KM-Plotter online tool databases, which offer anonymous data.

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
