# Peer review of "MYH9 Facilitates Cell Invasion and Radioresistance in Head and Neck Cancer via Modulation of Cellular ROS Levels by Activating the MAPK-Nrf2-GCLC Pathway"

_cells, 2022, doi:10.3390/cells11182855_

Round 1

Reviewer 1 Report

You et al. employed the use of shRNA against MYH9 approach to delineate the molecular mechanism of how MYH9 regulates anti-apoptotic, DNA damage response, resistance of HNC cells to apoptosis and balance of intracellular ROS level. The authors utilised 2 HNC cell lines, OECM-1 and Detroid 562 as models. The clinical aspects indicating the role of MYH9 in  HNC was shown using data pulled from TCGA. While the data seems convincing, however, there are several concerns, as listed below: 

1. It is a major concern that in all the figures, the authors use TCGA data. Authors should perform laboratory evidence to show the direct link of MYH9 to the expression of all target genes and proteins. Not just TCGA data. 

2. Technically, the authors did not mention clearly how many times the assays have been repeated. 

3. Figure 1A and B. Please show the efficiency of MYH9 knockdown for both migration and invasion assays. How many times have this assays been repeated? It was unclear if the Supplementary Fig. 1 result were directly for these 2 assays. 

4. Figure 3C. Which cell line is this mRNA result for? 

5. Section: Introduction, Line 58 - 59. How would the head and neck, rich in lymphatic tissues result in HNC?

6. The authors did not show that the MYH9 is overexpressed in the HNC cell lines tested in comparison to normal HNC cells. And whether MYH9 is mutated in HNC cells. If it is, what is the frequency of the mutation?

7. The authors missed out the picture that TP53 mutation, which is regulated by MYH9, could also affect how the expression of anti-apoptotic genes, DNA damage pathway and intracellular ROS level. 

Reviewer 2 Report

This manuscript studies the correlation between MYH9 expression, cancer agressiveness and radioresistance in 2 different head and neck cancer cell lines. Mechanistic causes are further explored.

The main conclusions are that MYH9 expression is associated with cell migration and invasion, radioresistance and reduced intracellular ROS levels, probably through regulation of antioxydant pathways.

Overall, the paper is well written, easy to read, with rational incremental steps in the described experiments. The authors can be congratulated for this in vitro work.

Some suggestions :

- as this "only" considers in vitro data for the moment, I would temper the final conclusion in the abstract, speaking about the need of further in vivo data to further assess the potential role of MYH9 as stratifying or therapeutic target.

- In the methods, regarding "cell lines" : is MYH9 overexpressed in these cell lines ?

- in the methods, regarding ROS levels : was there as usual a normalisation of flow cytometry results for number of cells ?

- Lines 179-180 are absolutely not clear. Please rephrase and precisely explain the "optimization algorithm". Readers should be able to perform the same analysis.

- Please mention the number of times the in vitro experiments were performed.

- was it always the same pool of cells with silenced MYH9 that was used for the experiments ? Or was this cell state renewed for each experiment ? Was the success rate of shRNA verified each time ?

- Line 235 : "may increase in" : this is causation, while authors study correlation. Please rephrase as in lines 202-203. Causation should be studied by qRT-PCR and WB and further experiments. Same in lines 282-284, this is only studied further in the manuscript.

- line 258 : was there a difference in % of living cells between the different ROS levels conditions ?

- Line 281, and also globally : please also show your negative/non significant results !

- Line 368 : in my opinion, this sentence requires to demonstrate that GCLC silencing does not lead a modification in MYH9 expression.

- Lines 433-448 : authors adequately mention WNT pathway. I would ask them to provide data regarding this pathway, in parallel to the already studied MAPK and AKT pathway : several papers link WNT with radioresistance and modification of extra-cellular matrix.

Finally, some sentence checks :

- line 35 : "was related to anti-apoptotic" : missing the word "expression of".

- line 129 : "will contain" ?

- line 135 : the cells were then "be" exposed.

- line 203 : ECM abreviation used for the first time, not worded in full

- line 276 : to up-regulate"d"

- line 328 : the suppressive effects ON gclc OF myh9

- line 359 : sentence to remove.

Author Response

We thank the comments for helping to make this manuscript a more meaningful document. Please see the attachment

Reviewer 3 Report

The submitted paper contributes to the understanding of the function of myosin heavy chain 9 (MHC9) in head and neck cancer. Authors suggest that MHC9 regulates cellular ROS levels via activating the MAPK-Nrf2-GCLC signaling pathway.

Comments

General comment

The function of MYH9 in the in vitro assays is interpreted from data from MYH9 silencing via shRNA. Do the authors condiser this as MYH9 is directly involved in cell migration, radioresistance, etc. or the silencing might also result in some indirect effects? 

Introduction

A hypothesis might be generated based on previous literature on MYH9, which is missing from the introduction.

Please also explain the use of the chosen cell lines, how these cells are relevant models for HNC. The use of a primary and a metastatic cell line is definitely a good choice. Also please mention that the findings in this paper are limited to oral/oropharynx localisations or might be considered more general to all localisations of the HNC. 

Methods

The methods are based on published references and described in sufficient details. 

One comment to the TCGA analysis. The gene expression and survival analysis covers all patients data from the TCGA or inclusion / exclusion criteria were also used? This also means were primary, secondary tumors, all localisations, HPV-positive and negative cases, etc. included? 

Results

The Results are described and mentioned in appropriate details. 

Discussion

The Discussion follows the Results and is well written. 

Author Response

(The authors gave the same response as above.)

Round 2

Reviewer 1 Report

You and colleagues provided substantial modifications to the manuscript, entitled "MYH9 facilitates cell invasion and radioresistance in head and neck cancer via modulation of cellular of ROS levels by activating the MAPK-Nrf2-GCLC pathway."  Below is my comments: 

Point 1: Including TCGA data certainly strengthen your observation. Authors should provide experimental evidence showing the change of expression of FN1, ITGA6, FSCN1, VIM, MMP2 and MMP9 genes in both malignant and normal head and neck cell lines used in this study. 

Point 7. I mean MYH9 regulates the TP53 stability. In most HNC cells, TP53 contains multiple mutations. TP53 expression also regulates multiple downstream transactivation genes involved in regulating apoptosis, DNA damage response and ROS level. Perhaps, the authors should discuss/comment/speculate on the role of MYH9 in this context.